# Analysis of the potential for point-of-care test to enable individualised treatment of infections caused by antimicrobial-resistant and susceptible strains of *Neisseria gonorrhoeae*: a modelling study

Katy ME Turner,[1] Hannah Christensen,[2] Elisabeth J Adams,[3] David McAdams,[4] Helen Fifer,[5] Anthony McDonnell,[6] Neil Woodford[5,6]

For numbered affiliations see end of article.

**Correspondence to**
Dr. Katy ME Turner;
katy.turner@bristol.ac.uk

## ABSTRACT

**Objective** To create a mathematical model to investigate the treatment impact and economic implications of introducing an antimicrobial resistance point-of-care test (AMR POCT) for gonorrhoea as a way of extending the life of current last-line treatments.

**Design** Modelling study.

**Setting** England.

**Population** Patients accessing sexual health services.

**Interventions** Incremental impact of introducing a hypothetical AMR POCT that could detect susceptibility to previous first-line antibiotics, for example, ciprofloxacin or penicillin, so that patients are given more tailored treatment, compared with the current situation where all patients are given therapy with ceftriaxone and azithromycin. The hypothetical intervention was assessed using a mathematical model developed in Excel. The model included initial and follow-up attendances, loss to follow-up, use of standard or tailored treatment, time taken to treatment and the costs of testing and treatment.

**Main outcome measures** Number of doses of ceftriaxone saved, mean time to most appropriate treatment, mean number of visits per (infected) patient, number of patients lost to follow-up and total cost of testing.

**Results** In the current situation, an estimated 33 431 ceftriaxone treatments are administered annually and 792 gonococcal infections remain untreated due to loss to follow-up. The use of an AMR POCT for ciprofloxacin could reduce these ceftriaxone treatments by 66%, and for an AMR POCT for penicillin by 79%. The mean time for patients receiving an antibiotic treatment is reduced by 2 days in scenarios including POCT and no positive patients remain untreated through eliminating loss to follow-up. Such POCTs are estimated to add £34 million to testing costs, but this does not take into account reductions in costs of repeat attendances and the reuse of older, cheaper antimicrobials.

**Conclusions** The introduction of AMR POCT could allow clinicians to discern between the majority of gonorrhoea-positive patients with strains that could be treated with older, previously abandoned first-line treatments, and

---

### Strengths and limitations of this study

► This study uses a simple framework to evaluate the potential impact of point-of-care tests to diagnose antimicrobial-resistant or antimicrobial-sensitive gonorrhoea infections.

► This study is parameterised with contemporary UK data on diagnoses, treatment and levels of antimicrobial resistance.

► This study uses a static model, so not possible to extrapolate future population effects.

---

those requiring our current last-line dual therapy. Such tests could extend the useful life of dual ceftriaxone and azithromycin therapy, thus pushing back the time when gonorrhoea may become untreatable.

## INTRODUCTION

Increasing antimicrobial-resistant gonorrhoea represents a significant and urgent public health problem. Gonorrhoea, caused by *Neisseria gonorrhoeae,* is the second most commonly diagnosed bacterial sexually transmitted infection (STI) in England. *N gonorrhoeae* has evolved resistance to all major drug classes and has been recognised as a bacterium of international concern by WHO[1] and has been prioritised in the UK 5-year antimicrobial resistance (AMR) strategy.[2]

Diagnoses have more than doubled from 16 839 in 2010 to 41 193 in 2015, mainly due to increased diagnoses in men who have sex with men (MSM), accounting for 70% of male infections in 2015, illustrated in figure 1[3] (data reported through GUMCADv2, including genitourinary medicine (GUM) clinics and other sexual health service providers, but

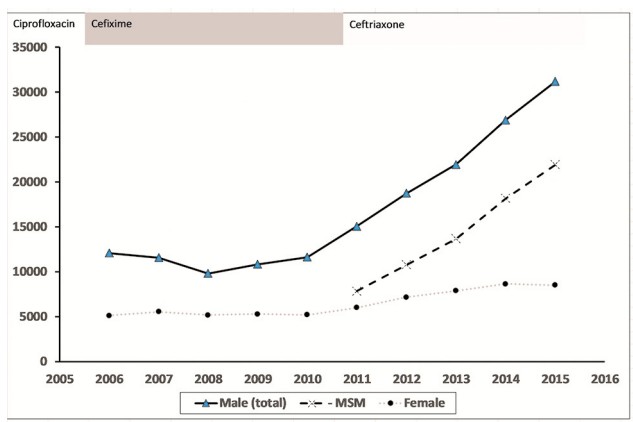

**Figure 1** Number of gonorrhoea diagnoses reported in England, 2006–2015, with the change in recommended first-line antibiotic treatment shown. Data from Public Health England, Annual STI Data Tables (https://www.gov.uk/government/statistics/sexually-transmitted-infections-stis-annual-data-tables). MSM, men who have sex with men.

not general practice). Infections are often asymptomatic, especially in women and in pharyngeal and rectal infections in MSM, but are still transmissible.[4] If untreated, complications of infection include pelvic inflammatory disease, infertility, increased risk of pregnancy complications and, in rare cases, life-threatening septicaemia.[5] Gonorrhoea infection also increases the risk of HIV acquisition.[6]

In the UK, the Gonococcal Resistance to Antimicrobial Surveillance Program (GRASP) has performed sentinel antibiotic susceptibility testing of gonorrhoea since 2000.[7] Increases in resistance to first-line therapies resulted in

two changes in treatment recommendation (figure 1): from ciprofloxacin to cefixime in 2005 and then to ceftriaxone plus azithromycin in 2011.[7–9] Our current first-line therapy is also our last-line option, and while the use of dual therapy is intended to delay resistance developing to ceftriaxone, decreased susceptibility to either of these drugs could lead to untreatable infections. While new antibiotics are in development, their use in the clinic may be many years away and already the world's first reported clinical treatment failure with confirmed ceftriaxone and azithromycin resistance has occurred.[7]

There are two main challenges to the management of gonorrhoea which contribute to the problem of resistance, illustrated in figure 2. (1) Precautionary treatment: at the time of diagnosis all infections are treated as if they are resistant to older antibiotics; and (2) epidemiological treatment: sexual contacts of gonorrhoea cases are often treated before diagnostic test results are known, resulting in unnecessary treatment of uninfected partners. The cornerstone of gonorrhoea management to date has been to ensure rapid, highly effective treatment is given to prevent the onward spread of infection to sexual partners and to prevent people not returning for treatment following a diagnosis. In the context of antibiotic resistance and new diagnostic technologies, it is necessary to reassess these priorities.

Strategies are required to extend the life of existing antimicrobials for the successful treatment of gonorrhoea. Most infections diagnosed in the UK are susceptible to cefixime, ciprofloxacin and even penicillin.[7] Therefore, if a point-of-care test (POCT) could be developed to test for resistance (or susceptibility) to antibiotics, most

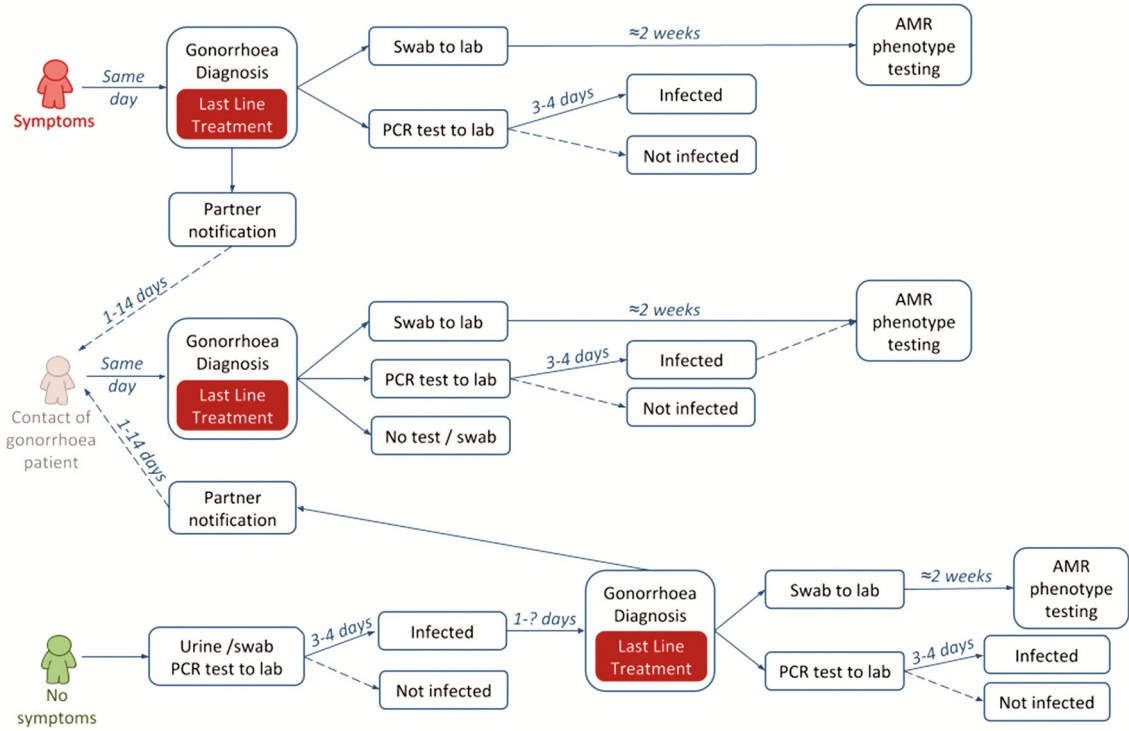

**Figure 2** Current patient pathways for gonorrhoea. AMR, antimicrobial resistance.

patients could be treated with an older oral first-line therapy, potentially extending the life of ceftriaxone as our last-line therapy.[10] A promising option based on existing nucleic acid amplification test (NAAT) could be a PCR test for ciprofloxacin resistance, using the gyrA gene as a target.[10 11] Other technologies could involve direct measurement of live cell responses to the presence of a panel of antibiotics including microfluidic devices, atomic force microscopy, volatile chemical detection or mass spectroscopy. Computational approaches based on in silico phenotyping based on genotype may also be able to detect new mutations more rapidly than traditional microbiological testing.[12–14] In this study, we developed a mathematical model to investigate the treatment impact and economic implications of introducing an AMR POCT for gonorrhoea.

## METHODS

### Model

We developed a decision tree model in Excel to consider the impact of a hypothetical new AMR POCT on testing, diagnosis and treatment of gonorrhoea in sexual health clinics in England (figure 3) compared with current practice. Genitourinary clinics typically triage attending patients based on whether they have symptoms or report

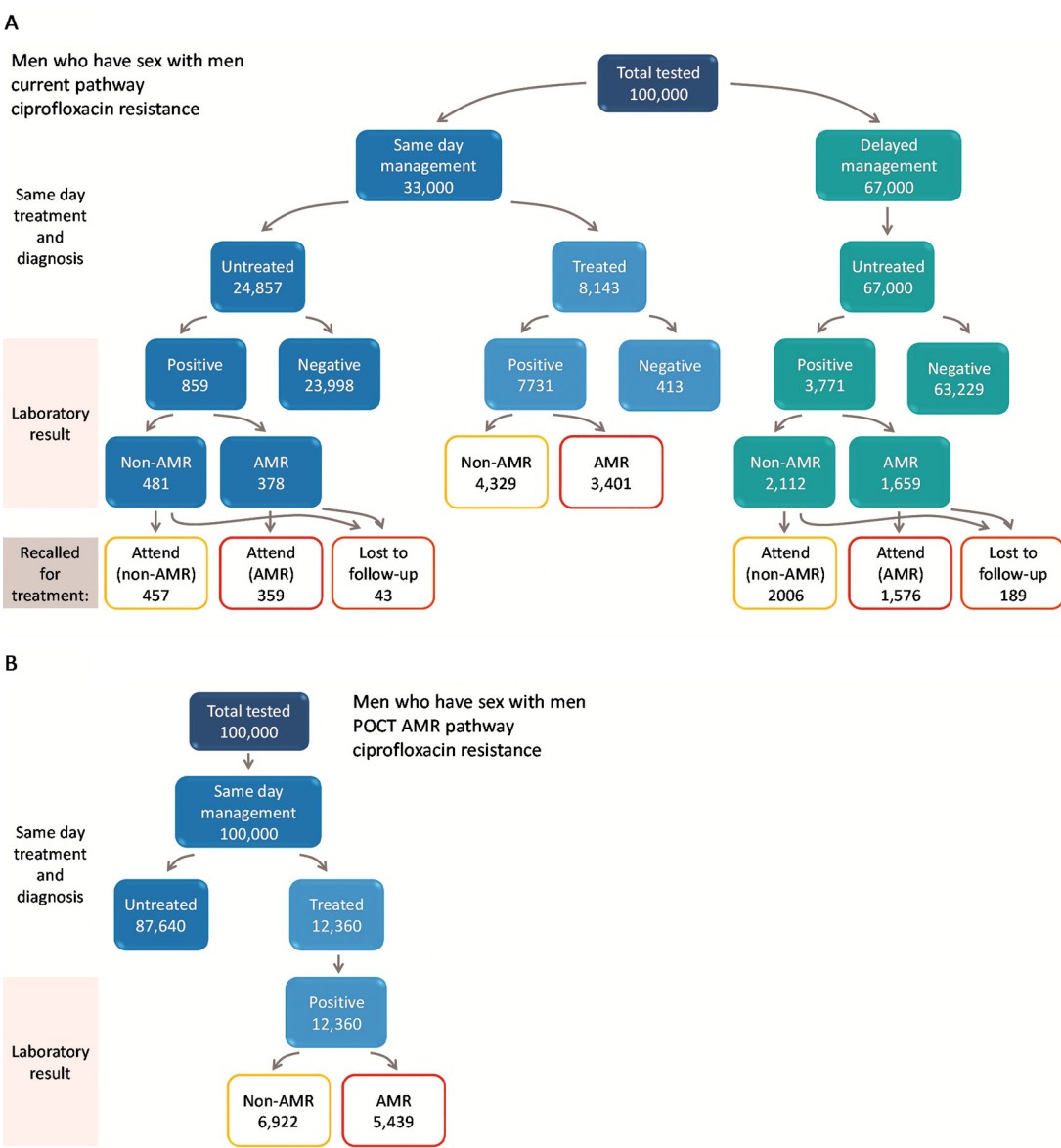

**Figure 3** Patient pathway diagram to illustrate the flow for men who have sex with men (MSM) under (A) current care and (B) antimicrobial resistance (AMR) point-of-care test (POCT). In scenario A, all diagnosed cases are treated with ceftriaxone plus azithromycin. In scenario B, diagnosed cases are treated according to resistance profile: AMR cases with ceftriaxone plus azithromycin; non-AMR with ciprofloxacin. Numbers of AMR and non-AMR infection are based on current levels of ciprofloxacin resistance observed in Gonococcal Resistance to Antimicrobial Surveillance Program surveillance data, 2014. Illustrated based on 100 000 MSM attending a genitourinary medicine clinic.

contact with a sexual partner infected with a specific infection ('same day management') and those without symptoms ('delayed management') where treatment is delayed until the results of diagnostic tests are returned from the laboratory (2–7 days) (figure 2). Current practice is therefore a mixture of same day management and delayed management depending on clinic patient mix. Guidelines recommend that patients treated for gonorrhoea also have swabs taken at the time of treatment that are sent for susceptibility testing, but these results are not available until after treatment has been given. The alternative strategy is based on a point-of-care gonorrhoea diagnostic test for all patients. The POCT could be either a simple diagnostic for gonorrhoea (infected/not infected) or a test which can discriminate between one specific resistance/susceptibility determinants (POCT AMR). Simple POCT tests are commercially available and have been piloted in clinic[15] but POCT AMR tests are still in development. More complex testing algorithms and diagnostic technologies could be envisioned, for example, only using an AMR POCT if the initial simple POCT is positive (reflex testing) or using more complex algorithms and new technologies to determine optimal treatment options. In this preliminary example, we consider two options of antimicrobial susceptibility: (1) ciprofloxacin and (2) penicillin.[16 17]

The model was based on an existing pathway model used to investigate the impact of introducing a dual POCT for gonorrhoea and chlamydia in a GUM setting,[17 18] but simplified in that onward transmission of gonorrhoea and partner notification were not included, with the focus being on diagnosis and tailored treatment, shown in figure 3 for MSM patient group (corresponding pathways for heterosexual men and women are given in the online supplementary appendix figure A1A-D). We explicitly included branches to differentiate susceptible and resistant isolates within the pathway framework. For the purpose of our study, we assumed that all POCTs have equivalent sensitivity and specificity to current PCR laboratory tests. Previous models have considered variable specificity and sensitivity requirements in more detail.[16]

Hypothetical cohorts of patients were followed through the pathway (MSM, heterosexual men and heterosexual women). Individuals could either receive same day management or delayed management (figure 3) under current practice or for POCT pathway all patients are assumed tested, diagnosed and treated on the same day. The only difference between POCT and AMR POCT is therefore in the choice of antimicrobial therapy. Treatments modelled were either our current last-line dual therapy of ceftriaxone and azithromycin (current pathway or simple POCT) or in the case of scenarios including AMR POCT a proportion of patients were provided with either ciprofloxacin or penicillin, plus azithromycin co-therapy, as an alternative regimen where possible. Loss to follow-up when patients were recalled for treatment following laboratory testing to determine positivity for gonorrhoea was explicitly included for current pathway

only. We assumed that results of point-of-care diagnostics can be provided within the clinical consultation, for example, if patients provide samples for testing on arrival at a GUM clinic and then wait for an appointment or return later in the day. It is possible that this would result in delays to treatment for symptomatic individuals and sexual contacts, but we do not consider this further.

## Parameter values

Full model parameters are provided in the online supplementary appendix tables A1 and A2. Estimates of the numbers of patients attending GUM clinics and tested for and diagnosed with gonorrhoea were based on recent data from Public Health England (PHE).[19] The model is run assuming 515 094 MSW, 145 863 MSM and 779 085 women attend a GUM clinic in 2014[19] and the proportions entering same day management or who are infected adjusted to generate the observed diagnoses of gonorrhoea in each group. In 2014, there were >33 000 diagnoses of gonorrhoea reported by PHE, just over half in MSM and the remaining heterosexual cases split roughly equally between men and women. We combined data on patients presenting as contacts of gonorrhoea cases or with symptoms into the 'same day management' pathway. Asymptomatic patients were tested, but treatment was assumed to be delayed until the results of laboratory tests were known. We distributed infected patients between the pathways according to specific parameters for each patient group based on the probability of being infected and the likelihood of having symptoms. Symptomatic patients are more likely to be managed on the same day as testing and heterosexual men (MSW) are the most likely to be symptomatic, followed by MSM, then women. (Data from the Maximising STI Control trial, personal communication Cath Mercer) (table 1).[17 18 20] These parameters were informed by national PHE data where available and supplemented with additional data or clinical experience and are described fully elsewhere.[17 20] The difference between MSM and MSW may be due to a combination of factors including higher probability of extragenital infection, higher incidence of repeat infections and higher probability of HIV coinfection and higher frequency of STI testing in this group.[21] We estimated the proportions of infections that are resistant to ciprofloxacin and/or penicillin from the GRASP 2014 report (online supplementary appendix table A1), which included systematic susceptibility testing at the PHE reference laboratory from sentinel surveillance sites and a larger but less well-defined analysis of samples tested locally.[22] Parameters were varied to be appropriate to three patient groups: heterosexual men, MSM and women. In the baseline case, we assumed that all confirmed and presumptive gonorrhoea infections are treated with ceftriaxone and azithromycin because there is >5% resistance to alternative regimens, resulting in 100% of infections treated as if they are resistant to other antibiotics (such as ciprofloxacin). The cost for patients attending GUM was taken from the latest payment by

**Table 1** Principal results comparing use of an antimicrobial resistance point-of-care test (AMR POCT) for ciprofloxacin (scenario 3a) or penicillin resistance (scenario 3b) against current testing practice (standard laboratory testing, no POCT) for the management of gonorrhoea (scenario 1), assuming the current attendance at genitourinary medicine clinic annually

| | Heterosexual male | MSM | Female | Overall |
|---|---|---|---|---|
| **Considering use of POCT test for ciprofloxacin resistance** | | | | |
| Annual ceftriaxone treatments | | | | |
| Current (scenario 1) | 7690 | 17 691 | 8050 | 33 431 |
| AMR POCT (scenario 3a) | 2188 | 7933 | 1257 | 11 378 |
| Reduction under scenario 3a | 5502 | 9759 | 6793 | 22 054 |
| Percentage reduction in ceftriaxone | 72 | 55 | 84 | 66 |
| Proportion treated same day (%) | | | | |
| Current (scenario 1) | 68 | 63 | 21 | 54 |
| AMR POCT (scenario 3a) | 100 | 100 | 100 | 100 |
| Increase under scenario 3a | 32 | 37 | 79 | 46 |
| Mean time to treatment (days) | | | | |
| Current (scenario 1) | 1.5 | 1.8 | 3.9 | 2.2 |
| AMR POCT (scenario 3a) | 0.0 | 0.0 | 0.0 | 0.0 |
| Reduction under scenario 3a | 1.5 | 1.8 | 3.9 | 2.2 |
| Persons lost to follow-up (untreated) | | | | |
| Current (scenario 1) | 125 | 338 | 329 | 792 |
| AMR POCT (scenario 3a) | 0 | 0 | 0 | 0 |
| **Considering use of POCT test for penicillin resistance** | | | | |
| Annual ceftriaxone treatments* | | | | |
| Current (scenario 1) | 7690 | 17 691 | 8050 | 33 431 |
| AMR POCT (scenario 3b) | 1407 | 4688 | 838 | 6932 |
| Reduction under scenario 3b | 6283 | 13 004 | 7212 | 26 499 |
| Percentage reduction in ceftriaxone | 82 | 74 | 90 | 79 |

*All other outcomes same as for use of POCT for ciprofloxacin resistance. Results for strategy 2 not shown: equivalent to strategy 3 except for choice of antibiotic treatment. Results for 3b also equivalent to 3a for outcomes except reduction in ceftriaxone treatments.
MSM, men who have sex with men.

results tariff.[23] An AMR POCT is not currently available so we assumed conservatively that separate new tests for assessing resistance to either ciprofloxacin or penicillin would each incur an additional £25 testing cost, similar to that previously assumed for a PCR-based POCT test.[17]

### Management scenarios

We considered the following scenarios for each of the three patient groups (MSM, heterosexual men and women).

1. Current management: Clinicians have no knowledge of the resistance profile of gonorrhoea at the point of initial treatment and consequently all patients are treated with ceftriaxone and azithromycin. Some patients are managed on the same day, either due to symptoms and positive microscopy or as contacts of infected individuals, others wait for lab results, resulting in some unnecessary treatment and some delays to treatment or loss to follow-up (figure 2).

2. Simple POCT management: All patients tested and managed same day but all treated as if resistant to older antibiotics (ie, ceftriaxone and azithromycin).

3. AMR POCT management: All patients tested with AMR POCT for gonorrhoea that could identify infections that do not need to be treated with ceftriaxone

   a. assuming current ciprofloxacin resistance prevalence[22] (figure 2).
   b. assuming current penicillin resistance prevalence.[22]

### Economic analysis

The primary outcomes were the number of doses of ceftriaxone saved and the mean time to appropriate treatment. In addition, we calculated the average number of visits per person and per infected person, the total cost of testing and the number of patients lost to follow-up. In each case, we compared the incremental

benefit of an AMR POCT with current testing practice. Analyses were undertaken from the National Health Service perspective with costs measured in pounds sterling at 2014 prices.

## RESULTS

We modelled a snapshot of GUM attendance, gonorrhoea diagnosis and prevalence of resistance to ciprofloxacin and penicillin based on the situation in England, 2014[19]. Under current treatment guidelines for 1.4 million people attending GUM per year, we estimate 33 431 ceftriaxone treatments are currently administered annually and 792 gonococcal infections remain untreated due to loss to follow-up. In those receiving antibiotics, the mean time to treatment was estimated to be 2.2 days. Under current practice, 68% (MSW), 63% (MSM) and 21% (women) who are infected with gonorrhoea are treated on the same day as they attend. The mean number of attendances at clinic per infected person was 1.44. We estimated the total cost of current testing to be £196 million. If a POCT test is used (strategies 2–4), this enables same day testing and treatment, patients would only need to visit once, all infected individuals would be treated on the same day as the test and therefore no infected individuals would be lost to follow-up and left untreated.

The results for AMR POCT (strategies 3 and 4) and POCT (strategy 2) only differ by the choice of treatment regimen. If an AMR POCT for ciprofloxacin resistance were available (strategy 3a), we estimate its use could prevent 22 054 treatments of ceftriaxone annually (a 66% reduction) assuming the current levels of resistance to ciprofloxacin (37% of infections in 2014,[24] table 1). Similarly, an AMR POCT for penicillin resistance (strategy 3b) at the current levels of resistance (23% overall) could prevent 26 499 ceftriaxone treatments annually (a 79% reduction). Assuming an AMR POCT added £25 to the testing costs, we estimated the total cost of testing for each of the POCT scenarios to be £230 million, adding £34 million to the annual cost of testing (table 2).

## DISCUSSION

### Statement of principal findings

Our model estimates that 66% of the 33 431 ceftriaxone treatments given annually to individuals with gonorrhoea could be replaced by ciprofloxacin, thus extending the life of our current last-line treatment, if an AMR POCT for ciprofloxacin resistance was available. If an AMR POCT for penicillin was available, 79% of ceftriaxone treatments could be substituted with penicillin. The use of POCTs would mean a 2-day reduction in the time that people wait, on average, for appropriate treatment compared with current practice and such testing would prevent the approximately 800 positive individuals who remain untreated in the current system due to loss to follow-up. If AMR POCT added £25 to first-line testing costs, we estimate the use of such tests would increase current treatment and testing costs by £34 million annually. The outcomes related to same day diagnosis and treatment (reduced time to treatment and reduced follow-up) could be achieved by using a simple POCT, as previously considered.[17] The additional benefit of AMR POCT test is to enable tailored choice of antimicrobial treatment.

### Strengths and weaknesses of the study

Our model used recent published data on AMR levels, gonococcal incidence and current treatment and considered the impact of additional AMR POCT in distinct population groups, namely heterosexual men, MSM and females. The simplified model structure, which is available freely online, enables the parameters to be easily updated and the impact of different scenarios, in different settings, to be considered. We made the simplifying assumption that the cost of an AMR POCT would add £25 to the current tariff cost; however, in reality other current activities might be reduced or discontinued if an AMR POCT was available, such as testing, microscopy, culture and physical exams or reattendances, as well as reduced costs associated with reusing cheaper oral antibiotics. New DNA-based POCT technologies may be able to be combined to produce a multiplexed test, which may be more economically viable than the separate specific AMR tests we modelled here. Our cost estimates are therefore

Table 2 Cost of testing and treatment* when using an antimicrobial resistance point-of-care test (AMR POCT) for ciprofloxacin resistance (strategy 3a) compared with current practice

| | Heterosexual male | MSM | Female | Overall |
|---|---|---|---|---|
| **Annual cost of testing (£)** | | | | |
| Current | 69 784 517 | 20 358 694 | 105 826 467 | 195 969 677 |
| AMR POCT | 82 415 040 | 23 338 080 | 124 653 600 | 230 406 720 |
| Increased cost with AMR POCT | 12 630 523 | 2 979 386 | 18 827 133 | 34 437 043 |

*The model assumes that the additional cost of AMR POCT (£25) is simply added to the cost of attendance and is not offset by reductions in the number of gonorrhoea infections by reduced treatment costs (as some patients are treated with cheaper antibiotics),or by reduced use of other tests (such as microscopy or culture of all swabs).
MSM, men who have sex with men.

likely to be higher than in practice. New technologies are emerging that may be able to rapidly determine the bacterial response to a panel of potential antibiotics which would enable highly tailored therapy without the need to continuously monitor the efficacy of a test for resistance based on detecting DNA sequence, but for this preliminary exploration we selected a hypothetical AMR POCT test which could integrate with existing POCT technologies based on nucleic acid amplification.

The model did not capture the indirect effects of reduced transmission to partners or progression to complications, such as pelvic inflammatory disease and epididymitis. It also did not consider the longer-term effects of changing treatment strategy on the evolution of drug resistance over time in gonorrhoea infections.

### Strengths and weaknesses in relation to other studies, discussing important differences in results

To our knowledge, no one has specifically addressed the question of the added value of an AMR POCT to discriminate between susceptible and resistant strains to guide initial treatment decisions for gonorrhoea. Others have considered in detail the relative benefits of POCTs, balancing the need for fast results against cost and test performance.[16] Adams *et al* previously showed that a dual chlamydia/gonorrhoea point-of-care NAAT diagnostic test pathway could be cost-neutral or cost-saving compared with existing methods even though the test kit itself is more expensive.[17 18] We initially assumed that the POCT AMR is an additional test cost; however, it is probable that a multiplex PCR rapid test could be designed to include an AMR component which does not compromise the cost or performance of the basic gonorrhoea diagnostic. An alternative to improving diagnostics, treatment and surveillance is to develop a vaccine for gonorrhoea and to improve the uptake of other methods of prevention (such as condoms).[25 26] A gonorrhoea vaccine has proved elusive due to the rapidly changing surface antigens, but there may be some cross-reactivity with vaccines designed to protect against *Neisseria meningitidis*.[27]

The main weakness of our study is that it did not address the population-level impact of the introduction of such tests, but only considered a static situation.[24 25 28] Rapid whole-genome sequencing (within 24 hours) has been introduced to help guide treatment decisions for important nosocomial pathogens, notably methicillin-resistant *Staphylococcus aureus*,[14] but in a community walk-in clinic setting for a low-prevalence bacterial infection, such as gonorrhoea, a test needs to be relatively cheap and results available before the patient leaves the clinic. Our model did not include dynamic epidemiological or evolutionary processes, which change the prevalence and incidence of infection (and resistance) over time.[24] In reality, reintroduction of ciprofloxacin would likely increase the selection for resistance, which would negate some of the benefits of an AMR POCT. Similarly, reusing other drugs would also result in increases in resistance observed, including increasing selection for plasmids conferring multidrug resistance. Conversely, if point-of-care technology can reduce the time to treatment and reduce loss to follow-up sufficiently this might reduce the overall population prevalence, which would lead to a virtuous cycle of improved control and reduced transmission risk.[29] We also assume that results of point-of-care diagnostics can be provided within the clinical consultation. This is not currently possible unless the patient provides samples on arrival then waits to see a clinician or returns for a later appointment. The Cepheid GeneXpert has a turnaround time of about 90 min which was previously found to result in the majority of men (16/19) not waiting for their results (six were positive).[30] Transmission dynamic models can explore the potential consequences without the risks associated with radical changes in prescribing practices. The next steps will be to develop dynamic models which include selective pressure under differing treatment options[31] and incorporating variable delays.

The important next questions arising from this study are: how much time does the reduction in use of ceftriaxone buy in terms of slowing or preventing the emergence of clinically relevant gonorrhoea resistant to ceftriaxone and, second, what are the population-level benefits of improved gonorrhoea control?

### Meaning of the study: possible explanations and implications for clinicians and policymakers

The major benefit of POCTs for gonorrhoea is increasing the proportion of patients treated appropriately on the same day as the test, which is likely to improve outcomes by reducing infectious duration, reducing loss to follow-up and potentially improving partner notification efficacy. A definitive diagnosis on the day of first presentation also prevents unnecessary treatment of those not infected with gonorrhoea. The main benefit of an AMR POCT that can discriminate between susceptible and resistant infections is in enabling the reintroduction of abandoned first-line therapies. Reducing the use of antibiotics, especially of last-line therapies, is a key aim of the UK national strategy on AMR. For heterosexual men and MSM, a relatively large proportion of infections are already treated on the same day as testing, based on epidemiological, clinical or microbiological evidence (microscopy). However, this proportion is lower for women due to the higher percentage of asymptomatic infections and from poorer sensitivity of detection of gonorrhoea in endocervical and urethral smears. Although new POCTs are likely to be more expensive than existing tests, this would to some extent be offset by the reduction in further attendances and in the ability to reuse older, cheaper drugs. Given the low prevalence of gonorrhoea even in high-risk GUM attendees, the cost of treatment and reattendances is small in comparison with the cost of attendances for testing and diagnosis. If a new discriminatory AMR POCT test were prohibitively expensive for routine use, a combination of a standard point-of-care NAAT (eg, chlamydia/gonorrhoea) could be considered in conjunction with

a more specialised gonorrhoea AMR test, although the time implications of this for patients and clinicians would have to be carefully considered.

## Unanswered questions and future research

This estimation of the potential reduction in ceftriaxone use is the first step towards evaluating the long-term effects of such a reduction. Future research investigating how much the useful lifespan of ceftriaxone as a therapy for gonorrhoea is extended with particular reductions in ceftriaxone use would be valuable. In the context of the often slow and expensive new drug pipeline, there is also a question to be answered around the value placed on each additional year of ceftriaxone availability.

**Author affiliations**
[1]School of Veterinary Sciences, University of Bristol, Langford House, Bristol, UK
[2]School of Social and Community Medicine, University of Bristol, Oakfield House, Oakfield Grove, Bristol, UK
[3]Aquarius Population Health, London, UK
[4]Duke Fuqua School of Business, Durham, USA
[5]Bacteriology Reference Department, National Infection Service, Public Health England, London, UK
[6]The O'Neill Review on Antimicrobial Resistance, Wellcome Trust, London, UK

**Acknowledgement** The authors acknowledge Lord Jim O'Neill, chairman of the Review on Antimicrobial Resistance and the review team, for commissioning the study.

**Contributors** All authors were involved in the conception and design of the research. KT, EA and HC developed the models, following initial work by DM and NW and based on previous published work by EA and KT. KT and HC analysed the model results and all authors interpreted the results. HF and NW provided input into current clinical practice relating to AMR. KT, HC and NW wrote the first draft of the manuscript. All authors drafted the final version of the manuscript. All authors had full access to all of the data in the study and can take responsibility for the integrity of the data and the accuracy of the data analysis.

**Funding** KT was funded by BristolBridge (grant number EP/M027546/1) under the EPSRC Bridging the Gaps between the Engineering and Physical Sciences and Antimicrobial Resistance cross-council AMR initiative. The model was developed from published work previously funded by Aquarius Population Health.

**Disclaimer** The views expressed are those of the author(s) and not necessarily those of the NHS, the NIHR, the Department of Health or Public Health England.

**Competing interests** KT reports personal fees from Aquarius Population Health, other from WHO, grants from Guys and St Thomas Charity , outside the submitted work; and is an editor of Sexually Transmitted Infections. HC reports grants from NIHR, during the conduct of the study; other from Sanofi Pasteur, outside the submitted work. HC was funded by the NIHR Protection Research Unit (NIHR HPRU) in Evaluation of Interventions at the University of Bristol in partnership with Public Health England (PHE). EA reports no compensation for the submitted work, and grants from Cepheid, Atlas Genetics, St Georges University of London, Enigma Diagnostics and AstraZeneca, outside the submitted work. AM reports personal fees from Department of Health, non-financial support from Wellcome Trust, outside the submitted work. NW reports PHE's AMRHAI Reference Unit receiving financial support from Achaogen Allecra Antiinfectives , Amplex, AstraZeneca UK, Becton Dickinson Diagnostics, BSAC, Cepheid, Check-Points B.V., Cubist Pharmaceuticals, Department of Health, Enigma Diagnostics, Food Standards Agency, GlaxoSmithKline Services, Henry Stewart Talks, IHMA, Merck Sharpe & Dohme, Meiji Seika Kiasya, Momentum Biosciences, Nordic Pharma , Norgine Pharmaceuticals, Rempex Pharmaceuticals, Rokitan, Smith & Nephew UK , Trius Therapeutics, VenatoRx and Wockhardt, outside the submitted work.

**Patient consent** No patient data used in this study except nationally published aggregate figures from PHE.

**Provenance and peer review** Not commissioned; externally peer reviewed.

**Data sharing statement** Details of the model data inputs and other assumptions are provided in the methods and supporting parameters table. The model is

available from http://amr-review.org/file/429 and researchers interested in further details may contact the corresponding author at katy.turner@bristol.ac.uk.

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
