## [Reviewer comments · BMJ Open]

This paper was submitted to a another journal from BMJ but declined for publication following peer review. The authors addressed the reviewers' comments and submitted the revised paper to BMJ Open. The paper was subsequently accepted for publication at BMJ Open.

ARTICLE DETAILS

TITLE (PROVISIONAL)	Analysis of the potential for point-of-care test to enable individualised treatment of infections caused by antimicrobial-resistant and susceptible strains of Neisseria gonorrhoeae
AUTHORS	Turner, Katy; Christensen, Hannah; Adams, Elisabeth; McAdams, David; Fifer, Helen; McDonnell, Anthony; Woodford, Neil

VERSION 1 - REVIEW

REVIEWER	Gibbs, Jo University College London, Infection and Population Health
REVIEW RETURNED	13-Aug-2016

GENERAL COMMENTS	This is an assessment of the treatment and economic impact of the introduction of hypothetical antimicrobial point of care tests for gonorrhoea which detect ciprofloxacin and penicillin resistance. This has not been previously assessed and is therefore of interest to the journal. I am not a mathematical modeller and am unable to comment in detail on the methodology employed. However, as the journal's target audience is not mathematical modellers, what I can offer is the opinion of a sexual health & HIV clinical academic. Overall, I would like more reassurance that this model is not overly optimistic and would like the authors to address the following comments: 1. My main comment is that the authors do not seem to have taken into consideration the impact of sensitivity and specificity of the AMR POCT – is this assumed to be 100%?2. It isn't very clear to me how the authors assumed that this will work in practice. I believe that this paper would have benefited from the input from a Genito-urinary physician. My interpretation is that all patients presenting to a GU clinic have a POCT for CT/GC (irrespective of risk/whether they are symptomatic or not) and only those who test positive will have separate AMR POCTs for ciprofloxacin and penicillin. Although the authors have made brief reference to this, it is a major limitation that how long the AMR POCT takes to process and whether patients will be willing to wait
--

for this result has not been factored into this modelling. The Cepheid POCT for chlamydia and GC takes 90 minutes to process and a previous study of the impact of this test on patient management in a clinic setting found only 3/19 men waited for their results (6 were positive). (Harding-Esch et al. ISSTD 2015). If an initial POCT needs to be performed followed by an AMR POCT, then the length of time a patient has to wait will be extended.

3. Prior to the introduction of NAATs in routine care, even when we had antibiotic sensitivities available prior to treatment, and where the GC was sensitive to ciprofloxacin and/or penicillin, we still used the recommended first line treatment to avoid increased levels of resistance.

The authors mention that re-introduction of ciprofloxacin would likely have this effect. However, they do not mention that it could consequently lead to increased resistance to penicillin (if used) and azithromycin as well. This could be covered.

4. Only the heterosexual male pathway is illustrated (Figure 1, page 20) however, MSM account for 70% of gonorrhoea diagnoses in men (Health Protection Report Vol 10 No 22 – 8th July 2016).

5. Presumably multi-site infection, with the need for testing at multiple sites for AMR, has not yet been factored into this model.

Minor Comments

1. Introduction (page 6, lines 19-25): the way this sentence is phrased could be interpreted as meaning that empirical treatment at the time patients first present could increase opportunities for transmission. It would be helpful if this was clarified.
2. Methods (page 10, lines 4-6): the proportion of the patients treated on the same day would have been because of gram negative diplococci being seen on microscopy or because they attended as a contact of someone who had been diagnosed with GC. What do you mean by 'epidemiological signs'?
3. Discussion (page 15, lines 29-30): the proportion of women treated on the same day as testing is also lower because of the poor sensitivity of detection of gonorrhoea in endocervical and urethral smears from women.
4. References (page 23, line 23-26): reference number 5 is incorrect.
5. Table A1 (page 26, line 19) – define GRASP.
6. Tables A1 and A2 (page 26 & 27)– consistency with number of decimal places.
7. Table A2 (page 27, line 20/21). Proportion who attend for treatment after lab test result: What is the denominator for the patients in the current pathway (i.e. 95% of what) as, for example, 96% of heterosexual men are described as being treated on the same day and it is currently confusing. It would be helpful to refer to either delayed management pathway or reference Figure 1 (page 20).
8. Table A2 (page 27, line 25) – why is the baseline model

	parameter for AMR POCT described with an additional AMR ('AMR POCT AMR')? Why is the cost of the AMR POCT included in the current pathway?
--	--

REVIEWER	Pickles, Michael Imperial College, Infectious Disease Epidemiology
REVIEW RETURNED	02-Nov-2016

GENERAL COMMENTS	The authors use a simple spreadsheet model to estimate the reduction in the number of ceftriaxone treatments through introduction of a point-of-care antimicrobial resistance test for gonorrhoea in England. The study does not give any real explanation of why this number is of scientific importance, noting only that it is a 'first step' (page 15, line 52) and an improved dynamical model is also under review (page 14, line 42), and indeed it is noticeable that the abstract conclusions do not in any way refer to the abstract results. Below are general additional comments on this paper. Title – 'impact' suggests that the authors are looking at how the number of cases of antimicrobial-resistant gonorrhoea may reduce. Can the authors use a title that better reflects their study. Page 7 line 9 "most patients could be treated with an older oral first-line therapy which could potentially extend the life of ceftriaxone" – it seems that the fundamental premise of this paper is that reducing use of ceftriaxone will reduce evolution of strains resistant to this drug. This seems likely, but it is not inconceivable that strains that have developed resistance to one form of treatment could be better placed to evolve resistance to other drugs (e.g. if the drugs are related). Can the authors comment on this. Page 9 line 19 – "...previous study Turner" is a typo. Figure 1 - Why is the population 100,000 in this figure (with 1,517 positives) and how do these numbers relate to those in table 1? If there are 34,958 reported cases in England, does that mean that the model is assuming that about 22 million people are screened (or that a smaller group – the "initial population size" in table A2 - are screened something like 14 times a year?) Table 1 - why do the authors present only reduction in numbers of ceftriaxone treatments? Surely % decrease in ceftriaxone treatments is a more suitable primary outcome? Negative reductions in mean time to treatment are surely double negatives.
---

	Table 2 - at present this is a meaningless set of numbers. How does the assumption of £25 per test affect the costs? Is £25 a potentially realistic number? (and I note that it is different to the £50 used in one of the supplementary reports by the same author). Either provide a breakdown of the costs (how much of the increase is due to the cost of the new test) or a sensitivity analysis given a range of plausible costs. As it stands I cannot see a reason for presenting 'costs' in this paper at all, as the 'costs' are based on an assumed cost of a non-existent test and do not include any long-term benefits (for example how reduced prevalence of drug-resistant gonorrhoea prevents future costs) and a naive reader will just see that this test is extremely expensive and therefore should not be considered. Strategy 2 – I cannot see the point of strategy 2 – the authors themselves only give a footnote in Table 1 and don't mention it in their results section. Other questions: Model assumptions -  - Sensitivity and specificity of the POCT AMR - this is assumed 100%. Can the authors comment on how realistic that is? - Model sensitivity analysis: can the authors either comment on the sensitivity of their model to the input parameters, or else contextualise their results (for example if these parameters represent the gonorrhoea epidemic in England in 2014, say that this is the context of the results). - Number of gonorrhoea cases – has this remained fairly static in England in years other than 2014?
--	---

VERSION 1 – AUTHOR RESPONSE

Reviewers comments from submission to Sexually Transmitted Infections	
Reviewer: 1	
1. My main comment is that the authors do not seem to have taken into consideration the impact of sensitivity and specificity of the AMR POCT – is this assumed to be 100%?	We have assumed 100% sensitivity / specificity for simplicity. We have clarified this in the text in Methods section : paragraph 2, last line “ For the purpose of our study, we

	assume that all point of care tests are 100% sensitive and specific for simplicity. Previous models have considered variable specificity and sensitivity requirements in more detail {Vickerman, 2003 #14}. “
2. It isn't very clear to me how the authors assumed that this will work in practice. I believe that this paper would have benefited from the input from a Genito-urinary physician. My interpretation is that all patients presenting to a GU clinic have a POCT for CT/GC (irrespective of risk/whether they are symptomatic or not) and only those who test positive will have separate AMR POCTs for ciprofloxacin and penicillin. Although the authors have made brief reference to this, it is a major limitation that how long the AMR POCT takes to process and whether patients will be willing to wait for this result has not been factored into this modelling. The Cepheid POCT for chlamydia and GC takes 90 minutes to process and a previous study of the impact of this test on patient management in a clinic setting found only 3/19 men waited for their results (6 were positive). (Harding-Esch et al. ISSTDR 2015). If an initial POCT needs to be performed followed by an AMR POCT, then the length of time a patient has to wait will be extended.	It is true that introducing delays in treatment to wait for results of susceptibility testing might require re-arrangement of current arrangements for providing prescriptions. The trade-offs between preserving ceftriaxone against potential delays in treatment requires further consideration in the context of a transmission dynamic model. However we believe the problem of drug resistance is severe enough that such issues require full assessment. We have clarified in the text as follows: We assume that results of point of care diagnostics can be provide within the clinical consultation, e.g. if patients provide samples for testing on arrival at a GUM clinic and then wait for an appointment or return later in the day. It is possible that this would result in delays to treatment for symptomatic individuals and sexual contacts, but we do not consider this further here as we are exploring the potential of theoretical new tests.

	Methods section – para 3, last 2 lines And also raised this as a discussion point Discussion – Strengths/weaknesses compared with other studies, final para We also assume that results of point of care diagnostics can be provided within the clinical consultation. This is not currently possible unless the patient provides samples on arrival then waits or returns for an appointment later. The Cepheid GeneXpert has a turnaround time of about 90 minutes which was previously found to result in the majority of men (16/19) not waiting for their results (6 were positive)²⁶. Including the ref given (thank you)
3. Prior to the introduction of NAATs in routine care, even when we had antibiotic sensitivities available prior to treatment, and where the GC was sensitive to ciprofloxacin and/or penicillin, we still used the recommended first line treatment to avoid increased levels of resistance. The authors mention that re-introduction of ciprofloxacin would likely have this effect. However, they do not mention that it could consequently lead to increased resistance to penicillin (if used) and azithromycin as well. This could be covered.	We expand on the consequences of re-using older drugs. Discussion, section Strengths/weaknesses wrt other studies Para 2 Similarly re-using other drugs would also result in increases in resistance observed, including increasing selection for plasmids

	conferring multidrug resistance.
4. Only the heterosexual male pathway is illustrated (Figure 1, page 20) however, MSM account for 70% of gonorrhoea diagnoses in men (Health Protection Report Vol 10 No 22 – 8th July 2016).	We included MSM and women in the calculation of the total cost and total number of treatments given (each pathway considered separately, then summed) – this has been clarified in the text in the methods section – management strategies.
5. Presumably multi-site infection, with the need for testing at multiple sites for AMR, has not yet been factored into this model.	No this hasn't been factored in and may also increase costs especially for MSM – added a point to discussion
1. Introduction (page 6, lines 19-25): the way this sentence is phrased could be interpreted as meaning that empirical treatment at the time patients first present could increase opportunities for transmission. It would be helpful if this was clarified.	We have substantially reworded the introduction to provide greater clarity and this sentence has been deleted.
2. Methods (page 10, lines 4-6): the proportion of the patients treated on the same day would have been because of gram negative diplococci being seen on microscopy or because they attended as a contact of someone who had been diagnosed with GC. What do you mean by 'epidemiological signs'?	Sentence changed to: Methods: Management strategies "Some patients are managed on the same day, either due to symptoms and positive microscopy or as contacts of infected individuals, others wait for lab results, resulting in some unnecessary treatment and some delays to treatment or loss to follow-up."
3. Discussion (page 15, lines 29-30): the proportion of	Thank you – amended sentence

women treated on the same day as testing is also lower because of the poor sensitivity of detection of gonorrhoea in endocervical and urethral smears from women.	However, this proportion is lower for women due to the higher percentage of asymptomatic infections and from poorer sensitivity of detection of gonorrhoea in endocervical and urethral smears.
4. References (page 23, line 23-26): reference number 5 is incorrect.	This sentence was deleted during rewriting of the introduction and more appropriate referencing of current treatment guidelines and GRASP reports given instead.
5. Table A1 (page 26, line 19) – define GRASP.	Amended
6. Tables A1 and A2 (page 26 & 27)– consistency with number of decimal places.	These reflect the accuracy to which the numbers are known or given in the data.
7. Table A2 (page 27, line 20/21). Proportion who attend for treatment after lab test result: What is the denominator for the patients in the current pathway (i.e. 95% of what) as, for example, 96% of heterosexual men are described as being treated on the same day and it is currently confusing. It would be helpful to refer to either delayed management pathway or reference Figure 1 (page 20).	Clarified – this is the assumed proportion diagnosed through a lab test who are treated in current management this is those not treated on the same day as attending the clinic. Assumption.
8. Table A2 (page 27, line 25) – why is the baseline model parameter for AMR POCT described with an additional AMR ('AMR POCT AMR')? Why is the cost of the AMR POCT included in the current pathway?	This extra AMR has been deleted. The cost for AMR POCT is not included in the current pathway – these are baseline parameters for the model
Reviewer: 2	
I doubt whether this model is necessary to estimate the effect of using abandoned fist-line antibiotics. To say very simplistic: interpreting the current AMR surveillance data already gives an estimate of the proportion of patients that can be treated with an alternative antibiotic. In addition, the POCT tests do not exist and estimated costs are unsure. In the model, no sensitivity analysis has been performed: both sensitivity and specificity of the new POCT were	Yes this is all true – however we believe our analysis is useful in highlighting some of the current issues with gonorrhoea management and how new test technologies might play a role as well as pointing out the need for further analysis. Specificity/sensitivity – see previous

assumed to be 100%.	reviewer comment.
The use of culture to detect Neisseria gonorrhoeae is not mentioned in this manuscript. Using this, the patients with delayed management could also receive alternative treatment without the development of POCT.	Yes this is true, although current practice is mainly PCR NAAT tests with culture done after diagnosis.
Page 3, line 42: add “ceftriaxone” between these and treatment.	
Page 3, Lines 42-44: What if the POCT AMR could detect susceptibility to both ciprofloxacin and penicillin? it would be very interesting to see the proportion of ceftriaxone reduction combined for ciprofloxacin and penicillin.	We agree that multiplex or more complex tests to detect combinations of resistance/susceptibility would be interesting, however this study was designed around the most likely tests to be developed (ciprofloxacin is due to chromosomal, single point mutations so is the easiest to detect through PCR for example).
Page 3, Line 48: “no positive patients remain untreated”: I find this to optimistic. Because this hypothetical POCT AMR has extraordinary discriminatory characteristics, no patients leave the clinic untreated. I would not see this as a main finding but a very logical, hypothetical result.	We have added discussion of potential loss of patient during wait for results (see reviewer 1) in the discussion. Agree this is a limitation but this is intended to be a theoretical exercise to stimulate future research
Key messages, Page 5, lines 5-9: please add that the inability to discern resistant strains is about the fact that we are not possible to do this at the same time as the moment of sampling.	Clarification added  • Most strains of Neisseria gonorrhoeae in the UK are susceptible to older, abandoned first-line treatments, but characterisation of the resistance/susceptibility profiles of infection is not available at the time of diagnosis and treatment.
Page 6, line 9: please describe the source of the data that 34,958 cases were reported. Does this only concern sexual health clinic data or also data from other health care provider like GP's?	This sentence has been updated with 2016 report and clarified as requested: more details are provided in the Tables from PHE Diagnoses have more than doubled from 16,839 in 2010 to 41,193 in 2015, mainly due to increased diagnoses in men who have sex with men (MSM), accounting for 70% of male infections in 2015,

	illustrated in Figure 1³ (data reported through GUMCADv2, including GUM clinics and other sexual health service providers, but not GPs).
Page 6, line 19: the recommended moment of treatment depends both on a (presumptive) test result, an STI notification or certain symptoms. The current sentence implies that everyone can receive treatment. Please elaborate on this. In addition, I would suggest to move this sentence to line 34.	This introduction has been reworked and clarified including addition of Figure 2 which shows the options more clearly and a new paragraph
Page 6, line 58: please add a reference for the data on resistant ciprofloxacin/penicillin (20-40% resistant).	This has been deleted in the new introduction.
Page 7, line 3: I do not agree that doctors lack the means: they can wait until AMR results are available. In fact there are two problems: 1) there is not a good POCT for gonorrhoea available and 2) for those patients who are treated because of being notified, having symptoms, a positive gram etc., no AMR data is available.	This has been clarified
General comment on the introduction: as a reader I would like to have some background about the development of POCT and AMR POCT. What is the current status of these tests (do they exist or are they in development or is it purely hypothetical)?	We have added a little more context on potential technologies and tools for acquiring information on AMR profiles more rapidly. Introduction No such test currently exists. A promising option based on existing nucleic acid amplification test (NAAT) could be a PCR test for ciprofloxacin resistance as this is conferred in a single, chromosomal mutation¹⁰. Other technologies could involve direct measurement of live cell responses to the presence of a panel of antibiotics including microfluidic devices, atomic force microscopy, volatile chemical detection or mass spectroscopy. Computational approaches based on in silico phenotyping based on genotype may also be able to detect new mutations more rapidly than traditional

	microbiological testing ¹¹⁻¹³ .
Methods	
Page 7, line 27: In the title of Figure 1 the focus is on heterosexual males. Is this right? In the manuscript, the model is also about MSM and women.	Yes the numbers are illustrated for heterosexual males, but the same pathway is used with slightly different parameters for MSM and women.
Page 7, lines 36-38: I feel like I miss the option of a culture as initial screening tool for gonorrhoea. Is this not done anymore in the UK? Please elaborate on this.	Most testing is NAAT PCR. Microscopy is done in GUM clinics for symptomatic patients, then swabs sent for culture which would take a couple of weeks for AMR testing. This is clarified in Figure 2 (new figure).
Page 8, line 52: Please move the abbreviation "MSM" after "with men".	Done
Page 9, line 13: Please remove "men who have sex with men" and use only MSM here.	Done
Page 9, line 13-15: Sentence between brackets (MSTIC) confuses me. Is this a kind of reference or does it belong to the next sentence?	Clarified (see sentence below)
Page 9, line 19: "previous study Turner": do you mean the previous study of Turner?	Clarified Page 9, line 19 Symptomatic patients are more likely to be managed on the same day as testing and heterosexual men (MSW) are the most likely to be symptomatic, followed by MSM, then women. (Data from the Maximising STI Control trial, personal communication Cath Mercer) (Table 1) ^{14,15,18}
Page 9-10, lines 52-20: in the methods the different options are called strategies. However in Table 1 they are called scenarios. This confused me in the beginning. Please use the same word at both places.	Amended to scenarios in both places
Page 10, line 48: the mean time to treatment was 2.2 days. I would like to see the proportion of patients that in the current management received delayed treatment and after introduction of POCT test is treated at the same day. For this group the mean time to treatment is	This is given in table 1 Proportions of those infected who are treated same day in current strategy

also interesting to include in the manuscript: especially for this group the POCT test will have an added value.	MSW MSM Women 68% 63% 21% And we have added to the text
Page 10, line 50: the mean number of attendances was 1.44: I would expect this to be lower: in current management the majority (1032 out of 1517) of gonorrhoea positive patients do receive same day treatment. If I estimate that 1032 visit once and 485 twice, the maximum mean visits should be 2002: $2002/1517=1,32$.	The numbers you give are just for heterosexual men – the women are much more likely to have delayed treatment and require a second visit so overall the average is 1.44.
Page 11, lines 9-11: In this line is stated that with ciprofloxacin a 66% reduction in ceftriaxone treatment can be reached. If 37% of the infections are resistant to ciprofloxacin, why is the reduction in ceftriaxone not 63%? The same applies for penicillin on page 12.	The slight difference is because of loss to follow up assumed in the current pathway.
Page 12, line 27-28: 2 days reduction in waiting time for treatment: I would suggest to focus on the group that receives delayed management in scenario 1. See also “Page 10, line 48” above.	We do agree that the most benefit in terms of reduction in time to treatment are the groups currently in the delayed management arm, but we think that is appropriate to calculate the average change in waiting time overall.
Page 12, line 40: please remove “diagnostic”.	Amended
Page 13, lines 17-27: please make use of two sentences to make this long sentence more comprehensible.	Apologies – I couldn’t work out which line this referred to
Page 13, line 58: assumed instead of assume.	Amended throughout
Page 14, lines 7-15: I think this parts does not add very much to the manuscript and could be omitted.	Page numbers do not coincide with my copy – have worked back from partner notification comment. I think this refers to the brief comment on vaccination and other interventions. We think it is important to mention other methods for control of AMR aside from new diagnostic tests
Page 15, line 13: “could improve partner notification”: I am interested why a POCT test would improve PN? Is there any proof shown in literature?	Theoretically a POCT could improve PN if both partners are tested at the same time in the clinic or if the reduced delay enables better recall of recent partners or people are more

	motivated by same day discussion with a health care professional. I am not aware of any specific data on this however – toned down to “potentially improve partner notification”
Page 15, line 54: remove “by” in the sentence “Future research investigating by how”.	Amended
Figure 1, Difference between AMR and non-AMR is not clear. Is this based on the proportion of gonorrhoea tests were a culture has been performed including resistance testing?	This is now Figure 3 In the model we assume a percentage of infections are AMR based on the reported percentage in GRASP, given in supplementary Table A1. This is clarified in the title and we have also referred to Figure 3 in the management scenarios definition in methods.
Figure 1, Under A: this is about current care: it reads like with AMR ciprofloxacin is given as treatment. I thought only ceftriaxone was given.	The scenarios are based on current levels of ciprofloxacin resistance. Title and legend have been clarified Figure 3 Patient pathway diagram to illustrate the flow for heterosexual males under A) current care, B) antimicrobial resistance point-of-care test Legend: In scenario A, all diagnosed cases are treated with ceftriaxone plus azithromycin. In scenario B, diagnosed cases are treated according to resistance profile: AMR cases with ceftriaxone plus azithromycin; non-AMR with ciprofloxacin. Numbers of AMR and non-AMR infection are based on current levels of ciprofloxacin resistance observed in GRASP surveillance data, 2014. Illustrated based on 100,000 heterosexual men attending a genitourinary medicine clinic.
Under B: the number of untreated (98438) and treated	This is a typo – thank you for spotting – it should be 98483 and 1517. This

(1517) do not add up to 100,000.	has been corrected
Table 1, Please add percentages to the row “Reduction under scenario 3a” for the annual ceftriaxone treatments.	Added in as requested and for penicillin for consistency
Reviewer: 3	
The authors use a simple spreadsheet model to estimate the reduction in the number of ceftriaxone treatments through introduction of a point-of-care antimicrobial resistance test for gonorrhoea in England. The study does not give any real explanation of why this number is of scientific importance, noting only that it is a ‘first step’ (page 15, line 52) and an improved dynamical model is also under review (page 14, line 42), and indeed it is noticeable that the abstract conclusions do not in any way refer to the abstract results.	We have clarified the scientific importance of the study in the introduction first paragraph
Title – ‘impact’ suggests that the authors are looking at how the number of cases of antimicrobial-resistant gonorrhoea may reduce. Can the authors use a title that better reflects their study.	Title changed Analysis of the potential for point-of-care test to enable individualised treatment of infections caused by antimicrobial-resistant and susceptible strains of Neisseria gonorrhoeae
Page 7 line 9 “most patients could be treated with an older oral first-line therapy which could potentially extend the life of ceftriaxone”	
– it seems that the fundamental premise of this paper is that reducing use of ceftriaxone will reduce evolution of strains resistant to this drug. This seems likely, but it is not inconceivable that strains that have developed resistance to one form of treatment could be better placed to evolve resistance to other drugs (e.g. if the drugs are related). Can the authors comment on this.	We have added a comment on evolution of multidrug resistance (see review 1 response)
Page 9 line 19 – “...previous study Turner” is a typo.	Amended
Figure 1 - Why is the population 100,000 in this figure (with 1,517 positives) and how do these numbers relate to those in table 1? If there are 34,958 reported cases in England, does that mean that the model is assuming that about 22 million people are screened (or	This is just an illustration of the patient pathway and it’s easier to interpret % of 100,000. The legend has been clarified. The model is run based on the observed number of GUM

that a smaller group – the “initial population size” in table A2 - are screened something like 14 times a year?)	attendances and the observed number of gonorrhoea cases in 2014. Sentence added to Methods: Parameter values, 1st para The model is run assuming 515,094 MSW, 145,863 MSM and 779,085 women attend a GUM clinic in 2014)¹⁷ and the proportions entering same day management or who are infected adjusted to generate the observed diagnoses of gonorrhoea in each group.
Table 1 - why do the authors present only reduction in numbers of ceftriaxone treatments? Surely % decrease in ceftriaxone treatments is a more suitable primary outcome?	This has been added (see reviewer 2 response)
Negative reductions in mean time to treatment are surely double negatives.	Agree “-“ removed
Table 2 - at present this is a meaningless set of numbers. How does the assumption of £25 per test affect the costs? Is £25 a potentially realistic number? (and I note that it is different to the £50 used in one of the supplementary reports by the same author). Either provide a breakdown of the costs (how much of the increase is due to the cost of the new test) or a sensitivity analysis given a range of plausible costs. As it stands I cannot see a reason for presenting 'costs' in this paper at all, as the 'costs' are based on an assumed cost of a non-existent test and do not include any long-term benefits (for example how reduced prevalence of drug-resistant gonorrhoea prevents future costs) and a naive reader will just see that this test is extremely expensive and therefore should not be considered.	We used both £25 and £50 in supplementary reports. Our most up to date understanding is that a new test would have to cost in the order of £25 for widespread use, unless subsidised, e.g. by government. Although the test doesn't exist, many companies are working on additions to current POCT PCR based tests which detect specific genetic markers of resistance and are likely to have equivalent cost to existing tests of this type. We have edited table 2 to also reflect that we are considering a work case number, ignoring savings from treating fewer people and using cheaper antibiotics or fewer consultations
Strategy 2 – I cannot see the point of strategy 2 – the authors themselves only give a footnote in Table 1 and	We have included it for completeness and to illustrate that most of the

don't mention it in their results section.	benefits described can be achieved without the AMR POCT , but just with a POCT – We feel this is an important point to make
- Model assumptions, sensitivity and specificity of the POCT AMR - this is assumed 100%. Can the authors comment on how realistic that is?	We have assumed 100% specific and sensitive. Although this may not be realistic we assume that a new test would have to have at least equivalent performance to current NAAT tests to be implemented.
- Model sensitivity analysis: can the authors either comment on the sensitivity of their model to the input parameters, or else contextualise their results (for example if these parameters represent the gonorrhoea epidemic in England in 2014, say that this is the context of the results).	We have added context to the results and also to the method section. First line, Results We modelled a snapshot of GUM attendance, gonorrhoea diagnosis and prevalence of resistance to ciprofloxacin and penicillin based on the situation in England, 2014 ¹⁷ . Under current treatment guidelines for 1.4 million people attending GUM per year we estimate
- Model assumptions, number of gonorrhoea cases – has this remained fairly static in England in years other than 2014?	Sentence added to introduction – number of gonorrhoea cases has been rising, especially in MSM where levels of resistance tend to be higher.

VERSION 2 – REVIEW

REVIEWER	Michael Pickles Imperial College London, UK
REVIEW RETURNED	27-Feb-2017

GENERAL COMMENTS	In this study the authors use a simple spreadsheet model to quantify the benefits, including doses of ceftriaxone saved, of introducing different hypothetical point-of-care tests for gonorrhoea in England. Although the model used has limitations, in particular not including onwards transmission of gonorrhoea or any dynamics of drug resistance, this represents an important first step in quantifying the potential benefits of such point-of-care tests. Given the necessity of developing strategies to address gonorrhoea resistance to ceftriaxone and azithromycin, the benefits suggested
--

	by this simple model warrant further investigation of the subject. In general this paper is clearly written, and the modelling well described. The only point which remains somewhat unclear is the hypothesised cost of £25 per test. Can the authors supply a reference to support whether this would be a realistic figure, for example by comparison to other similar point-of-care tests.
--	--

REVIEWER	Jo Gibbs University College London, UK
REVIEW RETURNED	12-Mar-2017

GENERAL COMMENTS	This is an assessment of the treatment and economic impact of the introduction of hypothetical antimicrobial point of care tests for gonorrhoea which detect ciprofloxacin and penicillin resistance. This has not been previously assessed and is therefore of interest to the journal. I am not a mathematical modeller and am unable to comment in detail on the methodology employed. However, as the journal's target audience is not mathematical modellers, what I can offer is the opinion of a sexual health & HIV clinical academic. I have previously reviewed this article for STI and am pleased to see that some of my recommendations and comments have been acted upon. However, I still have some concerns about this model and would like the authors to address the following comments:  1. My main concern is that the authors have assumed that the AMR POCT will have a 100% sensitive and specificity. This is extremely unlikely in practice and is a major limitation which effects the validity of the results. Particularly in the context of the other assumptions that have been made. 2. As existing NAATs do not have 100% specificity, even if a sexual partner tested negative for gonorrhoea, it could be a false negative and in clinical practice we would still consider treating them. 3. Presumably multi-site infection, with the need for testing at multiple sites for AMR, has not yet been factored into this model. 4. Only the heterosexual male pathway is illustrated (Figure 1, page 20) however, MSM account for 70% of gonorrhoea diagnoses in men (Health Protection Report Vol 10 No 22 – 8th July 2016). 5. Table A2 – Why is the cost of the AMR POCT included in the current pathway?
---

VERSION 2 – AUTHOR RESPONSE

Reviewer: 1

Reviewer Name: Michael Pickles

Institution and Country: Imperial College London, UK

Competing Interests: None declared

In this study the authors use a simple spreadsheet model to quantify the benefits, including doses of ceftriaxone saved, of introducing different hypothetical point-of-care tests for gonorrhoea in England. Although the model used has limitations, in particular not including onwards transmission of gonorrhoea or any dynamics of drug resistance, this represents an important first step in quantifying the potential benefits of such point-of-care tests. Given the necessity of developing strategies to address gonorrhoea resistance to ceftriaxone and azithromycin, the benefits suggested by this simple model warrant further investigation of the subject.

In general this paper is clearly written, and the modelling well described. The only point which remains somewhat unclear is the hypothesised cost of £25 per test. Can the authors supply a reference to support whether this would be a realistic figure, for example by comparison to other similar point-of-care tests.

Author response – This was the price used in our previous published study based on advice from Cepheid for the PCR based test technology used in their cartridge system. This is approx. 2x the current lab test costs and we consider that a more expensive test would not be economically viable in current clinical commissioning climate. A similar TB test costs \$17 per cartridge with additional running costs for example. However it is an assumption and we have clarified this in the text with reference to our previous study.

Reviewer: 2

Reviewer Name: Jo Gibbs

Institution and Country: University College London, UK

Competing Interests: None declared

This is an assessment of the treatment and economic impact of the introduction of hypothetical antimicrobial point of care tests for gonorrhoea which detect ciprofloxacin and penicillin resistance. This has not been previously assessed and is therefore of interest to the journal. I am not a mathematical modeller and am unable to comment in detail on the methodology employed. However, as the journal's target audience is not mathematical modellers, what I can offer is the opinion of a sexual health & HIV clinical academic.

I have previously reviewed this article for STI and am pleased to see that some of my recommendations and comments have been acted upon. However, I have still have some concerns about this model and would like the authors to address the following comments:

1. My main concern is that the authors have assumed that the AMR POCT will have a 100% sensitive and specificity. This is extremely unlikely in practice and is a major limitation which effects the

validity of the results. Particularly in the context of the other assumptions that have been made.

Author response – thank you for this comment and apologies for the misunderstanding from previous revision which is incorrect – in fact we assume that there is no difference between the current lab tests and the new POCT tests (such that any false negatives or positives would be equally frequent in either scenario). This has now been clarified in the text.

2. As existing NAATs do not have 100% specificity, even if a sexual partner tested negative for gonorrhoea, it could be a false negative and in clinical practice we would still consider treating them.

This an important consideration and is the precautionary principle of current management of partners.

However it is also critical to note that not all partners of an infected individual are themselves infected. In fact, only between 20-50% of partners of gonorrhoea infection are found to be infected in a range of studies which have assessed this. Treatment of uninfected partners is therefore a major contribution to use of last line ceftriaxone in UK.

Hard to get exact figures but the following study in US demonstrates the principle:

Concurrent Sexually Transmitted Infections (STIs) in Sex Partners of Patients with Selected STIs: Implications for Patient-Delivered Partner Therapy

Joanne Stekler Laura Bachmann Rebecca M. Brotman Emily J. Erbedding Laura V. Lloyd Cornelis A. Rietmeijer H. Hunter Handsfield King K. Holmes Matthew R. Golden

Clin Infect Dis (2005) 40 (6): 787-793. DOI: <https://doi.org/10.1086/428043>

Published: 15 March 2005 Article history

Gonorrhoea as index & also gonorrhoea found in following % of partners:

Women partners 221/470 (47%)

Heterosexual male partners 78/263 (29.7%)

MSM partners 101/299 (33.8%)

We have included this through partners managed in the “same day pathway” who may not be infected based on clinic data on numbers infected and treated from the MSTIC study population. We agree that this could be explored more explicitly (which are currently doing using a dynamic model) but is beyond the scope of the current paper.

3. Presumably multi-site infection, with the need for testing at multiple sites for AMR, has not yet been factored into this model.

- No we have not factored in multisite infections in this simple model structure

4. Only the heterosexual male pathway is illustrated (Figure 1, page 20) however, MSM account for 70% of gonorrhoea diagnoses in men (Health Protection Report Vol 10 No 22 – 8th July 2016).

This figure was included for illustration of a particular patient group through the management pathways.

We have redrawn the figure with all 3 patient groups. We have used the MSM pathway in the main text and added females and heterosexual males to appendix Figure A1.

We have presented full results for women, heterosexual men and MSM in all tables and overall.

5. Table A2 – Why is the cost of the AMR POCT included in the current pathway?

Removed

VERSION 3 – REVIEW

REVIEWER	Jo Gibbs University College London, UK
REVIEW RETURNED	20-Apr-2017

GENERAL COMMENTS	Thank you for addressing my comments. I have only one minor comment: I think it would be helpful to clarify somewhere in the manuscript that you have not considered multi-site infection and the rationale for this (i.e. beyond the scope of this simple model).
--